# Cross-Sectional Association between Hypercholesterolemia and Knee Pain in the Elderly with Radiographic Knee Osteoarthritis: Data from the Korean National Health and Nutritional Examination Survey

**DOI:** 10.3390/jcm10050933

**Published:** 2021-03-01

**Authors:** Byung Woo Cho, Du Seong Kim, Hyuck Min Kwon, Ick Hwan Yang, Woo-Suk Lee, Kwan Kyu Park

**Affiliations:** 1Department of Orthopaedic Surgery, Severance Hospital, Yonsei University College of Medicine, 50-1 Yonsei-ro, Seodaemun-gu, Seoul 03722, Korea; chobw0704@yuhs.ac (B.W.C.); dawe2005@yuhs.ac (D.S.K.); hyuck7777@yuhs.ac (H.M.K.); ihyang@yuhs.ac (I.H.Y.); 2Department of Orthopaedic Surgery, Gangnam Severance Hospital, Yonsei University College of Medicine, 211 Eonju-ro, Gangnam-gu, Seoul 06273, Korea; wsleeos@yuhs.ac

**Keywords:** osteoarthritis, knee pain, hypercholesterolemia, metabolic disease

## Abstract

Few studies have reported the relationship between knee pain and hypercholesterolemia in the elderly population with osteoarthritis (OA), independent of other variables. The aim of this study was to reveal the association between knee pain and metabolic diseases including hypercholesterolemia using a large-scale cohort. A cross-sectional study was conducted using data from the Korea National Health and the Nutrition Examination Survey (KNHANES-V, VI-1; 2010–2013). Among the subjects aged ≥60 years, 7438 subjects (weighted number estimate = 35,524,307) who replied knee pain item and performed the simple radiographs of knee were enrolled. Using multivariable ordinal logistic regression analysis, variables affecting knee pain were identified, and the odds ratio (OR) was calculated. Of the 35,524,307 subjects, 10,630,836 (29.9%) subjects experienced knee pain. Overall, 20,290,421 subjects (56.3%) had radiographic OA, and 8,119,372 (40.0%) of them complained of knee pain. Multivariable ordinal logistic regression analysis showed that among the metabolic diseases, only hypercholesterolemia was positively correlated with knee pain in the OA group (OR 1.24; 95% Confidence Interval 1.02–1.52, *p* = 0.033). There were no metabolic diseases correlated with knee pain in the non-OA group. This large-scale study revealed that in the elderly, hypercholesterolemia was positively associated with knee pain independent of body mass index and other metabolic diseases in the OA group, but not in the non-OA group. These results will help in understanding the nature of arthritic pain, and may support the need for exploring the longitudinal associations.

## 1. Introduction

Osteoarthritis (OA) of the knee joint is a very common disease among the elderly, and the number of affected patients continues to increase [1]. Joint pain is the most common manifestation of OA; however, its clear mechanism still remains unknown. In general, arthritic knee pain shows an “acute on chronic” pattern similar to that of chronic obstructive pulmonary disease or gout, and the intensity, frequency, and duration all vary widely depending on the patients [2]. Even among patients with the same degree of bone destruction, some complain of severe pain that makes walking difficult, while others may continue their daily lives without pain. This discordance between knee pain and severity of OA has been reported in previous studies [3,4].

Factors affecting arthritic knee pain are roughly divided into peripheral and central mechanisms [5,6]. Among peripheral pathologies, sensitization due to mechanical factors and inflammatory mediators have been considered as a main cause of OA pain [7]. Numerous mediators including nerve growth factor, chemokines, cytokines, and inflammatory cells were reported to be related to nociceptor sensitization [8]. As a link between OA and metabolic diseases has been recently identified [9,10,11], metabolically triggered low-grade inflammation is also known to induce nociceptor sensitization [12,13]. In animal studies, hypercholesterolemia due to high fat diet also appeared to be a factor causing synovitis and OA pain. In an experimental study using mice, Choi et al. reported that cholesterol metabolites in chondrocytes induce synovitis [14]. In another experimental study using rabbits, Larrañaga-Vera et al. reported that hypercholesterolemia induced by a high fat diet causes synovial inflammation [15]. In the clinical field, however, the relationship between cholesterol and OA remains controversial [16,17,18,19]. In addition, to our knowledge, there were no studies that reported factors influencing knee pain in the elderly with OA using a large-scale national data. Therefore, the aim of this study was to reveal the association between hypercholesterolemia and knee pain in the elderly with OA using a large-scale cohort. We hypothesized that hypercholesterolemia and associated metabolic diseases may affect the knee pain in patients with advanced OA.

## 2. Materials and Methods

### 2.1. Subjects and Study Design

A cross-sectional study was conducted using data from the Korea National Health and Nutrition Examination Survey (KNHANES-V, VI-1; 2010-2013) organized by the Korea Centers for Disease Control and Prevention (KCDC). Two-stage sampling (complex sampling) was used in this study, and it reflects elements such as stratification, clustering, and weights. This survey was conducted under the approval of the Research Ethics Review Committee of the KCDC (IRB No 2010-02CON-21-C, 2011-02CON-06-C, 2012-01EXP-01-2C, 2013-07CON-03-4C). Every year, approximately 10,000 individuals from 3840 households in 192 regions of South Korea are extracted as probability samples, and a screening examination, health survey, and nutrition surveys are performed. The period set in this study was the only period in which both radiographic knee OA stage and self-reported knee pain were investigated.

### 2.2. Assessment of Radiographic Knee OA and Knee Pain

Subjects underwent standing knee anteroposterior and lateral (30° flexion) radiographs, and three radiologists evaluated them using the Kellgren–Lawrence (KL) grading system. The discrepancy within 1 grade for the same case was 95.18%, and the kappa coefficient was 0.74. Radiographic OA was defined as a KL grade of 2 or higher [20], which accounted for a total of 3596 subjects (weighted number estimate = 20,290,421). The radiographs were performed on both sides, and the KL grade of the severe side was recorded. Knee pain was defined as experiencing knee pain for ≥30 days in the last three months, and the pain level was evaluated using a numeric rating scale (NRS) of 0 to 10 points [21]. When the pain in both knees were different, the level of the severe side was recorded.

### 2.3. Diagnosis of Metabolic Diseases

The associated diseases in this survey are defined as follows: hypercholesterolemia was defined as total cholesterol level ≥240 mg/dL after fasting for ≥8 h or taking cholesterol-lowering drugs; hypo-HDL (high-density lipoprotein)-cholesterolemia was defined as HDL-cholesterol level <40 mg/dL after fasting for 8 h; hypertriglyceridemia was defined as triglyceride (TG) level of ≥200 mg/dL after fasting for >12 h; obesity was defined as body mass index (BMI) of ≥25 kg/m^2^; hypertension was defined as blood pressure ≥140/90 mmHg or taking hypertensive medications; and diabetes was defined as fasting blood glucose level of ≥126 mg/dL after fasting for ≥8 h, a previous diagnosis by a physician, or taking diabetic medication (including insulin).

Metabolic syndrome (MetS) was defined using the National Cholesterol Education Program Adult Treatment Panel III and abdominal obesity criteria of the Korean Society for the Study of Obesity, and it was diagnosed when three or more of the following criteria were met [22,23]: (1) waist circumference ≥ 90 cm in men or ≥85 cm in women; (2) TG level ≥150 mg/dL; (3) HDL-cholesterol < 40 mg/dL in men or <50 mg/dL in women; (4) blood pressure ≥ 130/85 mmHg or antihypertensive medication use; and (5) fasting glucose ≥ 100 mg/dL or current use of anti-diabetes medication.

### 2.4. Definition of Associated Factors

Physical activities were divided into three categories (low, moderate, and high) based on the International Physical Activity Questionnaire (IPAQ) [24,25]. Alcohol consumption was based on high-risk drinking (seven or more drinks/episode for men, five or more drinks/episode for women) and classified into three groups: low (no episodes of high-risk drinking), moderate (episode ≤1 per month), and excessive (more than the moderate group) [26]. Household income was classified into quartile, and educational attainment was classified into none, less than elementary school, elementary school, middle school, high school, college, university, and graduate school.

### 2.5. Statistical Analysis

All statistical analyses were performed using a method of complex sampling analysis and the weighted number estimates. Student’s *t*-test and Chi-square test were used to compare the characteristics and prevalence of associated disease between the two groups divided by the presence of knee pain. Cochran–Armitage trend test was used to compare the proportion of knee pain according to the KL grade. Multivariable ordinal logistic regression analysis was performed to determine the associated factors for knee pain and to calculate the odds ratios (ORs). The multivariable ordinal logistic regression analysis was performed using only the variables showing statistical significance in univariable ordinal logistic regression analysis to adjust possible confounders. Statistical analysis was performed using the SPSS software for Windows (version 25.0, SPSS, Chicago, IL, USA) and R software version 3.6.3 (R Foundation for Statistical Computing, Vienna, Austria). *p*-values < 0.05 were considered significant.

## 3. Results

Among the subjects aged ≥60 years, 7438 subjects (weighted number estimate = 35,524,307) who replied knee pain item and performed the simple radiographs of the knee were enrolled and divided into the radiographic OA group with 3596 subjects (weighted number estimate = 20,290,421) and the non-radiographic OA group with 3842 subjects (weighted number estimate = 15,233,886) (Figure 1).

### 3.1. Prevalence of Knee Pain

Of the 35,524,307 subjects, 10,630,836 (29.9%) subjects experienced knee pain. Overall, 20,290,421 subjects (56.3%) had radiographic OA, and 8,119,372 (40.0%) of them complained of knee pain (Figure 2A). Higher KL grade was correlated with a higher proportion of subjects with knee pain (Figure 2B).

### 3.2. Comparison between Subgroups Distinguished by the Presence of Knee Pain In Subjects with OA

Subjects with radiographic OA (KL grade ≥ 2) were divided into two subgroups according to the presence of knee joint pain to compare the characteristics and prevalence of associated diseases (knee pain group and control group). The proportion of female sex (*p* < 0.001), age (*p* < 0.001), BMI (*p* < 0.001) and waist circumference (*p* = 0.004) were higher in the knee pain group than in the control group. In addition, the proportion of rural residents and those with low household income was higher in the knee pain group (*p* = 0.007 and *p* < 0.001, respectively). Obesity (*p* < 0001), hypercholesterolemia (*p* = 0.005), and MetS (*p* = 0.002) were more prevalent in the knee pain group. The levels of alcohol consumption, household income, education attainment, physical activity and severity of OA showed significant differences between the two groups (*p* < 0.001, respectively). Of the lipid profiles, only total cholesterol showed a significant difference between the two groups (*p* = 0.025) (Table 1).

### 3.3. The Result of Multivariable Ordinal Logistic Regression Analysis

After dividing the groups according to the presence of radiographic knee OA, multivariable ordinal logistic regression analysis was performed using the NRS of knee pain as the dependent variable (Table 2). Among the metabolic diseases, only hypercholesterolemia showed positive correlation with knee pain in the OA group (OR 1.24; 95% Confidence Interval (CI) 1.02–1.52, *p* = 0.033). There were no metabolic diseases correlated with knee pain in the non-OA group.

### 3.4. Comparison of Pain Levels between the Subgroups of Hypercholesterolemia

Subjects of the OA group were divided into three subgroups based on the previous diagnosis of hypercholesterolemia and blood cholesterol level; normal group (not diagnosed, with blood cholesterol level < 240 mg/dL), controlled group (diagnosed, with blood cholesterol level < 240 mg/dL) and uncontrolled group (blood cholesterol level ≥ 240 mg/dL). There was no difference in the knee pain level between the controlled group (2.96 ± 0.20) and the uncontrolled group (3.11 ± 0.26) (*p* = 0.631), but the knee pain levels of these groups were higher than that of the normal group (2.41 ± 0.09) (*p* = 0.011 and *p* = 0.009, respectively) (Figure 3).

## 4. Discussion

The aim of this study was to reveal the association between hypercholesterolemia and arthritic knee pain in the elderly. Using KNHANES data, a positive association between hypercholesterolemia and knee pain was revealed in elderly people with OA independent of BMI and other metabolic diseases. However, in the non-OA group, metabolic diseases were not associated with knee pain.

Obesity and high BMI are well-known risk factors for OA, and adipokines and cytokines produced by adipose tissue have been reported to be associated with knee joint pain [12]. Obesity and hypercholesterolemia are clinically closely related, and using large-scale human data, our study showed that hypercholesterolemia affects the presence of OA knee pain independent of BMI and other variables in subjects with OA. The direct effect of cholesterol on knee pain has not been clearly revealed in humans, but the following hypotheses can be considered. First, an innate immunity and inflammation may be the cause. Hypercholesterolemia accumulates cholesterol in immune cells to cause an innate immune response and systemic inflammation [27]. As a result, nociceptor sensitization occurs, which can increase knee OA pain [8]. Second, metabolite of cholesterol may be the cause. A study by Choi et al. reported that hypercholesterolemia increases cholesterol influx into chondrocytes, and retinoic acid-related orphan receptors produced by cholesterol metabolism induce cartilage destruction and synovitis in mice experiments. In addition, they reported that cholesterol metabolites affect joint pain using the von Frey assay and hot-plate assay, which indirectly measures pain in animal experiments [14]. Third, neuronal degeneration may cause pain. In addition to nociceptive pain and peripheral sensitization, evidence for neuropathic components is increasing, and it has also been reported that neuropathic pain in the hip and knee OA can be up to 23% [28]. Hypercholesterolemia increases the level of total plasma lysophosphatidic acid (LPA) [29], and an animal study by McDougall et al. suggested that LPA may lead to neuropathic joint pain by causing demyelination and nerve damage [30]. Although the exact mechanism in humans is not yet well understood, it is thought that these various factors are working in combination.

Many studies using large data sets have shown an association between OA and MetS [31,32,33]. In our study, multivariable ordinal logistic regression analysis also showed a positive relationship between MetS and the severity of radiographic OA (Appendix A). However, when analyzing knee pain as a dependent variable, MetS did not show a statistical relationship, but hypercholesterolemia was related. Although these are closely related diseases, it still remains unclear why their effects on OA itself and pain are different. Since MetS does not include an item on low-density lipoprotein (LDL) and the total cholesterol level is affected by LDL, it can be assumed that the effect of the two diseases on knee pain is different because of the role of LDL. However, it cannot be confirmed with our research data, and further study is required.

Interestingly, in the non-OA group, metabolic diseases such as hypercholesterolemia and MetS did not appear to affect knee pain. Factors that could induce mechanical stress, such as the female sex [34], BMI [35], and lifestyle factors (household income [36], educational attainment [37]), were associated with knee pain regardless of OA, but hypercholesterolemia was associated with knee pain only in the OA group. Therefore, it is possible that the induction of pain caused by hypercholesterolemia is more prominent in the knee joint with OA.

When the hypercholesterolemia group was divided into two subgroups, the controlled group and the uncontrolled group, there was no difference in knee pain level. Although this survey does not provide details on the control method, the result was the same as previous studies showing that statin, the most commonly used drug for dyslipidemia, did not affect OA pain [17,19]. The reason for this is not yet clear, but Riddle et al. suggested that the therapeutic statin dose for cholesterol lowering could differ from the optimal dose for OA-related symptoms [19]. Therefore, further longitudinal studies are expected to include the target total cholesterol levels or drug doses that may affect arthritic knee pain. If the causal relationship between hypercholesterolemia and knee pain is revealed, as well as the proper drug dosage, then we can use cholesterol-lowering drugs as another means of pain control.

This study had some limitations. First, due to the cross-sectional nature of this study, the causality between variables is unclear. Therefore, prospective studies are required to clarify the causal relationship. Second, it is also unclear whether pain originates from the knee joint, since it is based on the subject’s opinion. Moreover, the design of this study could not clearly reflect the characteristics of intermittent arthritic knee pain. Third, this survey did not provide detailed information on the classification of hypercholesterolemia and the diagnosis of hyper-LDL-cholesterolemia. Despite these limitations, our study has the following strengths: first, it is the first study to focus on arthritic knee pain in the elderly using large-scale data; and second, the results of this study found a modifiable factor that can control pain, even in patients with advanced OA.

## 5. Conclusions

This large-scale study revealed that in the elderly, hypercholesterolemia was positively associated with knee pain independent of BMI and other metabolic diseases in the OA group, but not in the non-OA group. These results will help in understanding the nature of arthritic pain, and may support the need for exploring the longitudinal associations.

## Figures and Tables

**Figure 1 jcm-10-00933-f001:**
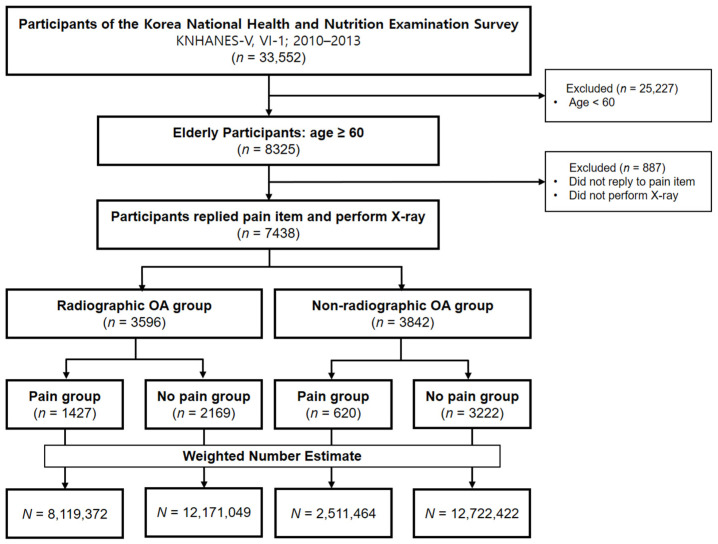
Flow chart of subject selection. OA: osteoarthritis.

**Figure 2 jcm-10-00933-f002:**
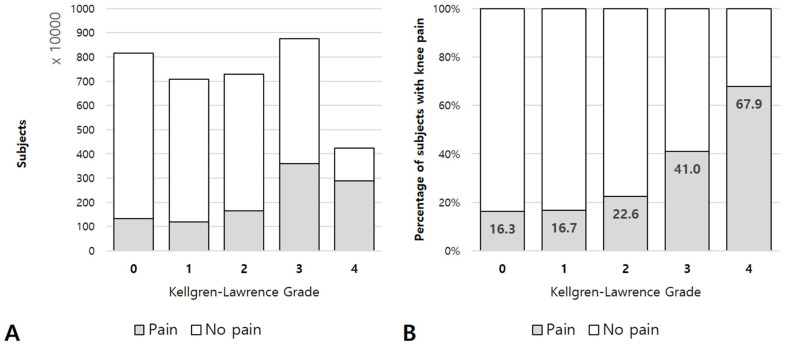
(**A**) Number of subjects according to Kellgren–Lawrence (KL) grades. (**B**) Percentage of subjects with knee pain for each KL grade. The proportion increased as KL grade increased (*p*-value for trend <0.001).

**Figure 3 jcm-10-00933-f003:**
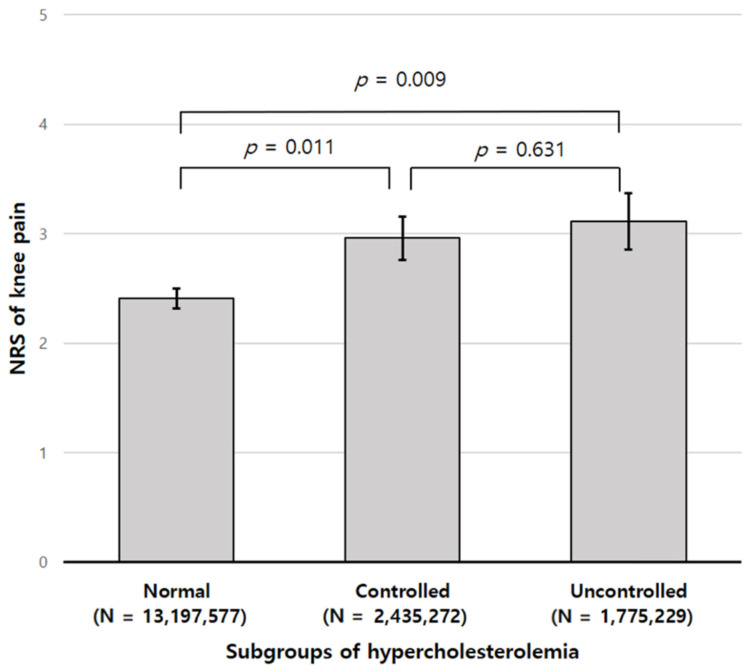
Comparison of mean numeric rating scale (NRS) of pain according to the subgroup of hypercholesterolemia. Values are presented as the mean ± standard error (SE).

**Table 1 jcm-10-00933-t001:** Demographic characteristics of two subgroups divided by the presence of knee pain among subjects with radiographic OA.

	Knee Pain Group (*n* = 8,119,372 ^†^)	Control Group (*n* = 12,171,049 ^†^)	*p*-Value
Female sex (%)	81.2	60.8	**<0.001**
Age (years)	71.5 ± 0.2	70.3 ± 0.2	**<0.001**
BMI (kg/m^2^)	25.0 ± 0.1	24.3 ± 0.1	**<0.001**
Waist circumference (cm)	85.7 ± 0.3	84.6 ± 0.2	**0.004**
Residential area (%)			**0.007**
Urban	63.3	68.7	
Rural	36.7	31.3	
Current smoking (%)	8.8	11.1	0.062
HTN (%)	66.6	63.1	0.050 ^‡^
DM (%)	23.4	20.7	0.124
Obesity (%)	46.8	39.1	**<0.001**
Metabolic syndrome (%)	53.9	47.6	**0.002**
Hypercholesterolemia (%)	27.2	22.3	**0.005**
Hypo-HDL-cholesterolemia (%)	25.9	25.5	0.856
Hypertriglyceridemia (%)	16.8	17.5	0.671
Lipid profiles			
Total cholesterol (mg/dL)	195.1 ± 1.2	191.6 ± 0.9	**0.025**
HDL-cholesterol (mg/dL)	47.5 ± 0.4	47.7 ± 0.3	0.669
Triglyceride (mg/dL)	144.6 ± 3.2	142.9 ± 2.3	0.670
LDL-cholesterol (mg/dL)	118.2 ± 3.3	114.5 ± 2.0	0.314
Alcohol consumption (%)			**<0.001**
Low	85.4	76.3	
Moderate	9.2	13.4	
Excessive	5.4	10.3	
Household income (%)			**<0.001**
Quartile 1 (low)	43.1	36.0	
Quartile 2	27.3	25.5	
Quartile 3	16.5	21.2	
Quartile 4 (high)	13.2	17.3	
Education attainment (%)			**<0.001**
Graduate school	0.2	1.2	
University	1.0	4.8	
College	0.3	1.2	
High school	7.0	17.6	
Middle school	13.5	16.5	
Elementary school	49.2	42.8	
Less than elementary school	26.9	14.5	
None	2.0	1.3	
Physical activity (%)			**<0.001**
Low	51.8	45.4	
Moderate	29.8	31.9	
High	18.4	22.7	
Kellgren–Lawrence grade			**<0.001**
Grade 2	20.3	46.4	
Grade 3	44.2	42.4	
Grade 4	35.5	11.2	

OA: osteoarthritis; BMI: body mass index; HTN: hypertension; DM: diabetes mellitus; HDL: high-density lipoprotein; LDL: low-density lipoprotein; Bold: *p* < 0.05. ^‡^
*p* > 0.05; Values are presented as the mean ± standard error (SE) or percent. ^†^ Number of subjects are presented as the weighted number estimate.

**Table 2 jcm-10-00933-t002:** Results of multivariable ordinal logistic regression analysis for knee pain (numeric rating scale) in the two subgroups divided according to the presence of radiographic OA.

Variables	OA Group (KL Grade ≥ 2) ^†^						Non-OA Group (KL Grade < 2) ^‡^					
	Univariable			Multivariable			Univariable			Multivariable		
	OR	95% CI	*p*-Value	OR	95% CI	*p*-Value	OR	95% CI	*p*-Value	OR	95% CI	*p*-Value
Metabolic syndrome	1.28	1.09–1.50	**0.002**	1.03	0.85–1.25	0.752	1.25	0.98–1.60	0.072			
Hypercholesterolemia	1.37	1.13–1.65	**0.001**	1.24	1.02–1.52	**0.033**	1.33	1.03–1.72	**0.027**	1.14	0.88–1.49	0.329
Hypo-HDL-cholesterolemia	0.98	0.82–1.17	0.831				0.98	0.75–1.29	0.909			
Hypertriglyceridemia	0.95	0.76–1.20	0.687				1.11	0.79–1.56	0.554			
Hypertension	1.16	1.00–1.34	0.055				1.10	0.89–1.36	0.382			
Diabetes	1.18	0.97–1.43	0.101				1.01	0.74–1.37	0.970			

OA: osteoarthritis; KL: Kellgren–Lawrence; OR: odds ratio; CI: confidence interval; HDL: high-density lipoprotein; Bold: *p* < 0.05. ^†^ Multivariable model was adjusted by sex, age, body mass index, residential area, household income, alcohol consumption, physical activity, education attainment and Kellgren–Lawrence grade. ^‡^ Multivariable model was adjusted by sex, age, body mass index, household income, alcohol consumption, current smoking, physical activity and education attainment. See Appendix A for detailed models.

## Data Availability

The data sets generated and/or analyzed during the current study are available on the KCDC homepage (http://knhanes.cdc.go.kr (6 March 2020)).

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
