# Peer review of "Cross-Sectional Association between Hypercholesterolemia and Knee Pain in the Elderly with Radiographic Knee Osteoarthritis: Data from the Korean National Health and Nutritional Examination Survey"

_jcm, 2021, doi:10.3390/jcm10050933_

Round 1
Reviewer 1 Report
BRIEF SUMMARY
The study aimed to establish, cross-sectionally, the association between knee pain and metabolic diseases in people with knee osteoarthritis (OA). Authors used existing data from a large cohort of people aged 60 years old and older and experiencing chronic knee pain. To investigate the association, authors conducted logistic regression models in a stratified group of people with radiographic knee OA and those without. Authors observed that hypercholesterolemia was positively associated with knee pain in people with radiographic knee OA, but not in people without radiographic knee OA.
I congratulate authors on their work. This is a well-written paper with informative figures and tables. The paper provides some useful information to better understand underpinning mechanisms of knee pain in people with knee OA. Overall, I found the topic timely and clinically important. My suggestions are listed below.
MAJOR COMMENT
My main concern is the quality of reporting. Please structure you article according to relevant reporting guidelines such as CONSORT or STROBE. Information on study design, inclusion/exclusion criteria, setting, recruitment procedures, statistical analysis is either superficially touched upon or lacking at the moment. Improving the quality of reporting will facilitate reading for both specialist and non-specialist audience.
SPECIFIC COMMENTS
TITLE
- Could authors include study design in the title, for example: “Cross-sectional association between….”
ABSTRACT
- Please include study aims in the abstract.
- “aged >60 years old” occurs unnecessarily twice.
- Lines 16-19. It gives an impression that having knee pain is an inclusion criterion (Line 16) but then you report that only 30% had knee pain (line 19). Please clarify to avoid confusion.
- Line 23: you introduce metabolic disease for the first time in the results. Please introduce earlier to avoid confusion.
- “may provide another treatment option when dealing with patients with OA”. I find this as overstatement given the cross-sectional nature of the study. At most, these results support the need for exploring the longitudinal associations or can be used to inform the design of more detailed prognostic models.
INTRODUCTION
- Lines 31-32. Reference is missing.
8.Lines 52-53. “In addition, to our knowledge, only a few studies have reported….”. First, please provide reference. Second, please explain then what is the added value of this study to the current knowledge.
- Please state your hypothesis at the end of the section, if you had one.
METHODS
- Please present information of study design and subjects in separate subsections.
- Line 64: Please provide reference number for the ethical approval.
- Please provide information on study inclusion and exclusion criteria in separate subsection.
- Line 77: “Radiographic OA was defined as a KL grade of 2 or higher”. Please provide reference
- There is no information whether you assessed both knees or just one, for both NRS pain and x-rays.
- Please add information regarding reliability and validity of the NRS pain and expand on the meaning of the scale.
- Line 81. Definition of variables. First please improve the structuring of this subsection. Were this outcomes, independent variables or covariates? This is currently not clear. Also, they are mentioned here for the first time in the paper which confuses the reader about the rationale of investigating them.
- Statistic. Please explain whether you checked normality of data and whether you used appropriate tests based on distribution of the data? Also, please explain how did you address the issue of multicollinearity in regression analysis. Finally, please add information regarding for which variables the models were adjusted for.
- There is no information on setting. Who assessed radiographic OA and knee pain and where, etc.?
RESULTS
- I suggest to move the flow chart here and carefully describe the study sample selection.
- I suggest authors to structure the results section into subsection as there appears to be done several subgroup analyses and in the current information flow it is difficult to figure out what exactly was done. The comparisons made should also be explained in the section on statistics.
DISCUSSION
- I suggest to start off the discussion with reminding the reader about the study objectives.
- Could authors also elaborate in the discussion on how the results can be exploited or inform future research?
Author Response
AUTHORS’ RESPONSE TO EDITOR and REVIEWERS’ COMMENTS
It is with pleasure that I resubmit to you a revised version of our manuscript. Thank you for your sincere and pertinent advice on this manuscript. We sincerely appreciate the time and effort you have invested to help us improve our paper. We have revised the manuscript and have provided a point-by-point response to your comments below.
MAJOR COMMENT
My main concern is the quality of reporting. Please structure you article according to relevant reporting guidelines such as CONSORT or STROBE. Information on study design, inclusion/exclusion criteria, setting, recruitment procedures, statistical analysis is either superficially touched upon or lacking at the moment. Improving the quality of reporting will facilitate reading for both specialist and non-specialist audience.
→ Thank you for your valuable comment. Our manuscript was modified according to the STROBE checklist for cross-sectional studies and your specific comments.
SPECIFIC COMMENTS
TITLE
- Could authors include study design in the title, for example: “Cross-sectional association between….”
→ Thank you for your valuable comment. It was modified according to your comment.
ABSTRACT
- Please include study aims in the abstract.
→ Thank you for your comment. It was added according to your comment.
- “aged >60 years old” occurs unnecessarily twice.
→ Thank you for your valuable comment. The latter one was deleted according to your comment.
- Lines 16-19. It gives an impression that having knee pain is an inclusion criterion (Line 16) but then you report that only 30% had knee pain (line 19). Please clarify to avoid confusion.
→ Thank you for your valuable comment. It was modified to avoid confusion.
- Line 23: you introduce metabolic disease for the first time in the results. Please introduce earlier to avoid confusion.
→ Thank you for your detailed comment. It was described together by adding an aim to the abstract.
- “may provide another treatment option when dealing with patients with OA”. I find this as overstatement given the cross-sectional nature of the study. At most, these results support the need for exploring the longitudinal associations or can be used to inform the design of more detailed prognostic models.
→ Thank you for your important comment. It was modified according to your comment.
INTRODUCTION
- Lines 31-32. Reference is missing.
→ Thank you for your comment. It was added according to your comment.
- Lines 52-53. “In addition, to our knowledge, only a few studies have reported….”. First, please provide reference. Second, please explain then what is the added value of this study to the current knowledge.
→ Thank you for your comment. The sentences were modified, and the added value of this study was described in the aim & hypothesis. As you may know, we emphasized the OA pain rather than the OA development.
- Please state your hypothesis at the end of the section, if you had one.
→ Thank you for your comment. It was added according to your comment.
METHODS
- Please present information of study design and subjects in separate subsections.
→ We described that in “Subjects and study design” subsection.
- Line 64: Please provide reference number for the ethical approval.
→ It was added according to your comment.
- Please provide information on study inclusion and exclusion criteria in separate subsection.
→ Thank you for your comment. It was added at the end of the subsection.
- Line 77: “Radiographic OA was defined as a KL grade of 2 or higher”. Please provide reference
→ Thank you for your comment. It was added according to your comment.
- There is no information whether you assessed both knees or just one, for both NRS pain and x-rays.
→ Thank you for your valuable comment. X-ray was performed on both sides, and the KL grade of the severe side was recorded. When both knee pain were different, the level of the severe side was recorded. These were added in manuscripts.
- Please add information regarding reliability and validity of the NRS pain and expand on the meaning of the scale.
→ Thank you for your comment. Since we used the NRS adopted in national surveys, it seems unnecessary to describe the rationale and meaning in our manuscript. Instead, associated citation was added.
- Line 81. Definition of variables. First please improve the structuring of this subsection. Were this outcomes, independent variables or covariates? This is currently not clear. Also, they are mentioned here for the first time in the paper which confuses the reader about the rationale of investigating them.
→ Thank you for your comment. We restructured the subsection and modified the end part of the introduction section according to your comments.
- Statistic. Please explain whether you checked normality of data and whether you used appropriate tests based on distribution of the data? Also, please explain how did you address the issue of multicollinearity in regression analysis. Finally, please add information regarding for which variables the models were adjusted for.
→ Thank you for your comment. The logistic regression we used is independent of the normality test. When checking multicollinearity in ordinal logistic regression, a variable that is not related to the result is set as a dependent variable, and multicollinearity between independent variables is checked through multiple linear regression analysis. The VIFs of all variables used in our model were less than 2. Information on the variables used for adjustment when performing multivariable regression is provided in the additional description of Table 2.
- There is no information on setting. Who assessed radiographic OA and knee pain and where, etc.?
→ Pain levels were evaluated by survey. The radiographic OA evaluation was conducted by three radiologists specializing in the musculoskeletal system.
RESULTS
- I suggest to move the flow chart here and carefully describe the study sample selection.
→ Thank you for your comment. It was modified according to your comment.
- I suggest authors to structure the results section into subsection as there appears to be done several subgroup analyses and in the current information flow it is difficult to figure out what exactly was done. The comparisons made should also be explained in the section on statistics.
→ Thank you for your valuable comment. It was modified according to your comment.
DISCUSSION
- I suggest to start off the discussion with reminding the reader about the study objectives.
→ Thank you for your comment. It was added according to your comment.
- Could authors also elaborate in the discussion on how the results can be exploited or inform future research?
→ Thank you for your important comment. It was modified according to your aforementioned comment to avoid overstatement.
Reviewer 2 Report
The present study aimed to investigate an association between knee osteoarthritis (KOA) pain and metabolic diseases including hypercholesterolemia. For this purpose, the authors conducted a cross-sectional study using data from a large cohort of the Korea National and Nutrition Examination Survey. Criteria inclusion for the analysis were presence/absence of radiographic KOA, age>60 years old and self-reported knee pain leading to a final sample size of 7438 subjects.
As a first view, the research is based in a vast number of patients being indicative of robustness and solid evidence. A brief background introduces the connection of metabolic disorder to OA by describing data from preclinical models to the uncertain clinical setting. The study design is correct and studied variables are well defined. Perhaps including raw data as supplementary material of such a large cohort would bring additional support and transparency to the investigation.
However, the following concerns might be clarified before publication:
- What do the authors mean with “All statistical analyses were performed using a method of complex sample analysis”? Can the authors clarify this statement?
- Can the authors explain what they meant by the weighted number estimates? What is the purpose of including it in the flow chat diagram of figure 1 that shows patient selection and enrollment in the study?
- Figure 2. What is the main title to describe this figure describing the relationship between knee pain and OA severity? 2B. Remarking several times the percentage unit in a graph is somehow redundant, please modify this issue.
- 121. “Higher K-L grade was correlated with higher proportion…” Can the authors comment the statistical test used to correlate knee pain with the K-L score?
- Table 1. No bold data is highlighted, so why including it in the table legend? A statistical significant difference (p=0,025) was found on the total cholesterol value with an absolute difference of about 3,5 mg/ml although a similar difference among the LDL (about 3,7 mg/ml) did not reach a p-value lower than 0,05 (p=0,314). Why? Was the data of the entire population available for each studied demographic characteristic? If not this might be stated in the text.
- 127 to 129. Results concerning sex, age, BMI and waist circumference are mentioned twice in both text and table 1 leading to repetitive information given in the manuscript.
- Figure 3: Numeric rating scale NRS should be cited in the figure caption. There is no need to add asterisk mean p<0,05 as p values are well described between the bars.
- 176, l.203, l.210. Citing tables and figures in the discussion section seem to be inadequate. This correspond to the results section.
- L198-201. Do the authors have additional explanation to the association found between knee pain and hypercholesterolemia but not with metabolic syndrome (MetS) as the first is included within the features of the MetS. The authors may specify the kind of further analysis required to confirm this point.
Author Response
AUTHORS’ RESPONSE TO EDITOR and REVIEWERS’ COMMENTS
It is with pleasure that I resubmit to you a revised version of our manuscript. Thank you for your sincere and pertinent advice on this manuscript. We sincerely appreciate the time and effort you have invested to help us improve our paper. We have revised the manuscript and have provided a point-by-point response to your comments below.
- What do the authors mean with “All statistical analyses were performed using a method of complex sample analysis”? Can the authors clarify this statement?
→ In order to improve the representativeness of the sample and the accuracy of estimation, the sampling area (survey district) of the KNHANES was extracted by the multi-level stratified cluster probability sampling, which is a complex sampling method. This is a concept in contrast to simple random sampling, generally used as an efficient method when sampling units are widely spread out geographically. The three elements of complex sampling design (weight, stratification and clustering) were included in the raw data DB and disclosed. When analyzing the data of the KNHANES, a method that considers complex sampling design information should be used. If this is not taken into account, biased results would be obtained from the estimates (mean, prevalence, odds ratio, variance and standard error)
- Can the authors explain what they meant by the weighted number estimates? What is the purpose of including it in the flow chat diagram of figure 1 that shows patient selection and enrollment in the study?
→ The weight of the KNHANES is an expanded multiplier given that the estimated value represents the entire population of South Korea, and is calculated by reflecting the extraction rate, response rate, and population distribution. We presented the flow chart to show the number of subjects and their weighted number estimate of each group.
- Figure 2. What is the main title to describe this figure describing the relationship between knee pain and OA severity? 2B. Remarking several times the percentage unit in a graph is somehow redundant, please modify this issue.
→ Thank you for your comment. It was modified according to your comment.
- “Higher K-L grade was correlated with higher proportion…” Can the authors comment the statistical test used to correlate knee pain with the K-L score?
→ Thank you for your comment. It was added according to your comment. Since the Cochran-Armitage trend test is not provided by SPSS, we also added information about R software.
- Table 1. No bold data is highlighted, so why including it in the table legend? A statistical significant difference (p=0,025) was found on the total cholesterol value with an absolute difference of about 3,5 mg/ml although a similar difference among the LDL (about 3,7 mg/ml) did not reach a p-value lower than 0,05 (p=0,314). Why? Was the data of the entire population available for each studied demographic characteristic? If not this might be stated in the text.
→ There seems to be a technical error after submission so we highlighted again.
The reason that the p values of total cholesterol and LDL cholesterol are different is that the characteristics of the variables are different.
Total cholesterol: mean estimate 192.96, standard error 0.735, 95% CI 191.51-194.40
LDL-cholesterol: mean estimate 115.91, standard error 1.815, 95% CI 112.34-119.48
- 127 to 129. Results concerning sex, age, BMI and waist circumference are mentioned twice in both text and table 1 leading to repetitive information given in the manuscript.
→ Thank you for your comment. It was modified to avoid repetition.
- Figure 3: Numeric rating scale NRS should be cited in the figure caption. There is no need to add asterisk mean p<0,05 as p values are well described between the bars.
→ Thank you for your comment. It was modified according to your comment.
- 176, l.203, l.210. Citing tables and figures in the discussion section seem to be inadequate. This correspond to the results section.
→ Thank you for your comment. It was deleted according to your comment.
- L198-201. Do the authors have additional explanation to the association found between knee pain and hypercholesterolemia but not with metabolic syndrome (MetS) as the first is included within the features of the MetS. The authors may specify the kind of further analysis required to confirm this point.
→ Thank you for your valuable comment. As you may well know, MetS includes low serum HLD and high serum TG, and total cholesterol level is determined by the level of LDL, HDL and TG. The reason that knee pain is associated with hypercholesterolemia and not MetS is presumed to be due to the role of LDL, but this cannot be confirmed due to data limitations. Therefore, further studies on the role of LDL are needed, and the according content was added in our manuscript.
Round 2
Reviewer 1 Report
I thank authors for their work. My comments have been well addressed.
Author Response
Thank you very much.